# Association of Colonic Diverticula with Colorectal Adenomas and Cancer

**DOI:** 10.3390/medicina57020108

**Published:** 2021-01-25

**Authors:** Angelo Viscido, Fabiana Ciccone, Filippo Vernia, Dolores Gabrieli, Annalisa Capannolo, Gianpiero Stefanelli, Stefano Necozione, Giorgio Valerii, Hassan Ashktorab, Giovanni Latella

**Affiliations:** 1Gastroenterology Unit, Department of Life, Health and Environmental Sciences, University of L’Aquila, 67100 L’Aquila, Italy; angelo.viscido@univaq.it (A.V.); filippo.vernia1@gmail.com (F.V.); annalisacap@tiscali.it (A.C.); giastefanelli@gmail.com (G.S.); 2Gastroenterology Unit, Giuseppe Mazzini Hospital, 64100 Teramo, Italy; fabiana.ciccone@hotmail.it (F.C.); doloresgabrieli@gmail.com (D.G.); gioval83@hotmail.it (G.V.); 3Epidemiology Unit, Department of Life, Health and Environmental Sciences, University of L’Aquila, 67100 L’Aquila, Italy; stefano.necozione@univaq.it; 4Department of Medicine and Cancer Center, Howard University College of Medicine, Washington, DC 20059, USA; hashktorab@howard.edu

**Keywords:** diverticulosis, diverticulitis, diverticular disease, colorectal polyps, colorectal adenoma, colorectal cancer

## Abstract

*Background and Objectives:* Conflicting evidence is reported regarding any association between colonic diverticula with colorectal adenomas or cancer. The present study aimed to evaluate, in a cohort of Caucasian patients, the association between colonic diverticula and colorectal polyps and cancer. *Materials and Methods:* All consecutive patients undergoing colonoscopy at our institution were included in the study. The presence and location of diverticula, polyps, and cancers were recorded. Histologically, polyps were classified as adenoma (with low or high dysplasia), hyperplastic, or inflammatory. The relative risk of the association of polyps and cancer with diverticula was assessed. Multiple logistic regression analyses, including age, sex, family history for colorectal cancer (CRC), and family history for diverticula, were carried out. *Results:* During the study period, 1490 patients were enrolled; 37.2% (*n* = 555) showed colonic diverticula or polyps or CRC (308 males, mean age 66 years). Particularly, 12.3% (*n* = 183) patients presented only diverticula, 13.7% (*n* = 204) only polyps or cancer, 11.3% (*n* = 168) both diseases, and 62.7% (*n* = 935) neither diverticula nor polyps and cancer. A total of 38 patients presented colorectal cancer, 17 of which had also diverticula. A significant increase in relative risk (RR 2.81, 95% CI 2.27–3.47, *p* < 0.0001) of colorectal adenoma and cancer in patients with colonic diverticula was found. At multivariate analysis, only diverticula resulted to be significantly associated with colorectal adenomas and cancer (Odds Ratio, OR 3.86, 95% CI 2.90–5.14, *p* < 0.0001). *Conclusions:* A significant association of colonic diverticula with colorectal adenoma or cancer was found. This implies that patients with colonic diverticula require a vigilant follow-up procedure for the prevention of colorectal cancer from those applicable to the general population.

## 1. Introduction

Colonic diverticula and colorectal polyps and cancer are common findings during colonoscopy, especially in patients over 50 years of age [1]. Most patients with diverticula of the colon remain asymptomatic throughout their lives, a condition commonly referred to as diverticulosis or asymptomatic diverticula. Approximately 20% of those with colonic diverticula develop symptoms (abdominal pain or discomfort, bloating, constipation, or diarrhea), conditions described under the general heading of symptomatic diverticular disease (DD), which may be uncomplicated, recurrent, or complicated [2,3,4]. Symptomatic DD includes the types without inflammation (symptomatic uncomplicated diverticular disease (SUDD)) and those with overt inflammation of the colon, including both segmental colitis associated with diverticula (SCAD) and diverticulitis [5]. This latter form can be acute or chronic, uncomplicated, or complicated.

Patients with colorectal polyps and early colorectal cancer (CRC) are, in most cases, asymptomatic [6,7]. Bowel habit alterations and/or hematochezia appear in a minority of cases and only in the presence of large and multiple polyps or ulcerated and/or stenotic CRC. The screening tests are aimed at detecting asymptomatic polyps and CRC [8,9]. A recent study demonstrated that bleeding-related signs (overt bleeding, iron deficiency anemia, or a positive fecal occult blood test) are present in three-quarters of patients with CRC. Both overt and occult bleeders are less likely than non-bleeders to have metastases at diagnosis. Overt bleeders are very unlikely to present with obstructive symptoms and most occult bleeders are found to have proximal cancer [10].

The prevalence of both diverticula and colorectal neoplasia increases exponentially with age.

Under the age of 40, the frequency of diverticula is estimated at approximately 5%, increasing to 65–70% for those over 65 [11].

CRC is a highly common malignancy in European countries and the world in general [12,13,14].

According to GLOBOCAN data, 1.36 million new cases, affecting 17.2 per 100,000 populations, are diagnosed worldwide each year and 693,000 people die from CRC, amounting to a yearly mortality rate of 8.4 per 100,000 [14]. CRC has become a major public health concern because, despite treatment, as many as half of CRC patients die from the disease [13]. CRC is sporadic in 90% of patients, whereas in <10% it is inherited [15,16] or can be the complication result of inflammatory bowel disease (IBD), either ulcerative colitis (UC) or Crohn’s disease (CD) [17,18,19,20]. In most cases, CRC develops from adenoma, a preclinical benign precursor, the progression from early adenoma to invasive cancer taking years [21]. The estimated CRC lifetime risk is 5–6% with incidence rates increasing sharply after the age of 50 [21,22].

In addition to age, diverticula and colorectal neoplasms appear to share other risk factors, such as a diet low in fibers and rich in saturated fats, obesity, and slow colonic transit time [1,2].

Inflammation also seems to be a common finding shared by the two conditions, the presence of a chronic inflammatory process increasing the risk of a malignant transformation [20,23,24]; chronic microscopic inflammation of the colon has been reported in DD, even in the SUDD [25].

All this evidence gives rise to the hypothesis of a possible association between diverticula and colorectal adenomas and cancer. However, contrasting results have been reported as some studies supporting this thesis [1,26,27,28,29], while others demonstrate an absence of this association [30,31,32]. Most of the evidence available in literature derives from retrospective studies, and the results from the few prospective cohorts are discordant, especially among different ethnic groups. Thus, further prospective studies are needed to establish the real relationship between DD, colorectal polyps, and CRC, as this could have important implications for CRC screening programs. The present study aimed to evaluate, in a cohort of Caucasian patients, the association between colonic diverticula, colorectal polyps, and cancer. 

## 2. Materials and Methods

### 2.1. Study Population

This prospective study was carried out at the Gastroenterology Unit of the Medical School of the University of L’Aquila, L’Aquila, Italy, and the Gastroenterology Unit, Giuseppe Mazzini Hospital, Teramo, Italy.

All consecutive patients who underwent colonoscopy in the period between September 2016 and September 2019 were included. Patients underwent the procedure for a variety of reasons: uncomplicated lower abdominal pain or discomfort, hematochezia, changes in bowel habit, weight loss, iron-deficiency anemia, chronic constipation, chronic diarrhea, surveillance after colonic polypectomy, and screening for CRC.

Ethics Committee Approval: The study was performed with the institutional review board’s approval (Prot. N. 43958). All clinical investigations were conducted according to the principles laid down in the Declaration of Helsinki.

Informed consent: Written informed consent was obtained from all patients included in the present study.

### 2.2. Colonoscopy and Histopathology

Before the colonoscopy, patients received a polyethylene glycol solution for bowel preparation and intravenous diazepam or midazolam for sedation. During the study period, 1994 colonoscopies were performed. A total of 504 patients did not meet the inclusion criteria, as 311 were affected by IBD, infectious colitis, or ischemic colitis, 137 had poor bowel cleansing (Boston score 0–1) [33], 54 had incomplete endoscopic examination, and 2 had polyps close to the scar, resulting from previous polypectomies. A total of 1490 patients were therefore enrolled and included in the statistical analysis. All enrolled patients were divided into four groups according to the endoscopic findings: group A, patients with only DD; group B, patients with only polyps or CRC; group C, patients showing both DD and polyps or CRC; and group D, patients without diverticula, polyps, or cancer (Figure 1). Data were recorded on a standard database.

Colonoscopy findings were documented using a standardized colonoscopy-reporting system. Colonic diverticula are characterized by the herniation of the colonic mucosa and submucosa through defects in the muscle layer at the weakest point in the colonic wall. Diverticulosis was defined as the presence of at least two diverticula, in the absence of symptoms. DD was defined as intestinal symptoms and signs related to the presence of colonic diverticula [5,34].

Colorectal polyps were classified according to their location and histopathology. We defined the proximal colon as the cecum, ascending colon, and transverse colon, including splenic flexure, distal colon as descending colon, sigmoid colon, and rectum. All polyps were removed with a forceps biopsy (polyps < 5 mm) or a diathermic loop (polyps > 5 mm), as appropriate. Multiple biopsies were taken from CRC.

Histological samples were classified as cancer, adenoma (with low or high dysplasia), hyperplastic, or inflammatory polyps.

### 2.3. Statistical Analysis

Comparisons among the groups were assessed by the chi-square test and Fisher’s exact test, as appropriate, for categorical variables. The relative risk (RR), with the 95% confidence intervals of the association of polyps and cancer with diverticula, was assessed. Multiple logistic regression analyses, including age, sex, family history for CRC, and family history for diverticula, were carried out. All statistical analyses were performed using the STATA version 15.1 2017 (StataCorp LLC, College Station, TX, USA).

## 3. Results

Of the 1490 patients enrolled, 37.2% (555/1490) showed colonic diverticula, polyps, or CRC (308 M, 247 F, mean age 66 years). In particular, 23.5% (351/1490) showed colonic diverticula, 15.0% (224/1490) colorectal adenomas, 2.5% (37/1490) CRC, 6.5% (97/1490) hyperplastic polyps, 0.9% (14/1490) inflammatory polyps, and 62.7% (935/1490) had other different lesions, such as proctitis, angiodysplasia, hemorrhoids, or no mucosal lesions. The study population was divided into four patient groups (Figure 1): Group A: 12.3% patients (183) with only diverticula (52% male vs. 48% female); Group B: 13.7% patients (204) with only polyps or cancers (53% male vs. 47% female); Group C: 11.3% patients (168) with both diverticula and polyps or cancers (62% male vs. 38% female); Group D: 62.7% patients (935) without neoplastic lesions or diverticula (56% male vs. 44% female).

Clinical characteristics of the four groups of patients are summarized in Table 1. Histological characteristics and localization of neoplastic lesions and diverticula of Group C (patients with both polyps or CRC and diverticula) are summarized in Table 2. In this group of patients, most polyps were found in the distal colon.

In particular, inflammatory polyps were only found in the distal colon. The histological distribution of all polyps and cancers found in Group B (patients with only polyps or cancer), in Group C (patients with both diverticula and polyps or cancers), and Group B + Group C is shown in Figure 2; the most frequent histological finding was adenomatous polyps, accounting for 60% (*n* = 224) of all lesions.

Out of all the 555 patients (Group A + Group B + Group C), 6.7% (37) of them showed a CRC; out of these, 45.9% (17) also showed diverticula. Adenomatous polyps were found in 40.4% (224/555) patients; out of these, 104 (45.4%) also showed diverticula. Hyperplastic polyps were found in 17.5% (97/555) patients; out of these, 38.1% (37) also showed diverticula. Inflammatory polyps were found in 2.5% (14/555) patients; out of these, 71.4% (10) also showed diverticula.

The RR of the association of polyps and cancer with diverticula was assessed. A significant increase in RR of the association of polyps and cancer with diverticula of the colon was found. The RR was higher when only adenomas and cancer were evaluated compared to all types of polyps and cancer (2.81 vs. 2.67, respectively) (Table 3 and Table 4).

At multivariate analysis (Table 5), of the demographic (age, sex) and clinical parameters (family history of colonic diverticula and neoplastic lesions) considered, only diverticula resulted to be significantly associated with colorectal adenomas and cancer (odds ratio (OR) 3.86, 95% CI 2.90–5.14, *p* < 0.0001).

## 4. Discussion

Colon cancer could be associated with a history of diverticular disease and personal history of polyps that may be detected during colonoscopy, especially in older patients (>50 years old).

Many studies have been conducted on colonic diverticula but they are heterogeneous due to the different study designs (prospective, retrospective, case-control), the different classes of patients considered (population-based, hospital-admitted, outpatients, etc.), the various methods employed for the diagnosis of diverticula (barium enema, computed tomography (CT), surgery, endoscopy), and the different diseases for which the evaluations were conducted (diverticular disease, diverticulitis, polyps, adenoma, cancer) (Table 6) [1,26,27,28,29,30,31,32,35,36,37,38,39,40,41,42,43,44,45,46,47,48,49,50,51,52,53,54,55,56,57,58,59,60,61,62,63,64,65,66,67,68,69,70].

Most of the evidence available in literature derives from retrospective studies. However, large prospective cohorts have been recently described in Asia [42,49,53,60,65]. Excluding data published in the form of an abstract, to our knowledge, only three prospective studies have been carried out in Caucasian patients [1,26,64] and two in mixed cohorts from the USA [32,41] (Table 6). The incidence and distribution of diverticula in the colon is different between Western countries and Asian countries. In the former, the diverticula are more frequent and develop more in the left colon, while in the latter they predominate in the right colon [2,3,4,5].

In this prospective study, including all consecutive Caucasian patients, we found a significant association of colonic diverticula with colorectal adenomas or cancer. Additionally, the majority of all polyps (including both inflammatory and adenomatous polyps) were found in the distal colon, where diverticula were also more frequently present.

A recent meta-analysis involving 29 cross-sectional studies reported that diverticula were associated with increased risk of adenomas (OR 1.47, 95% CI 1.18–1.84) and with the more lax-defined “polyps” (OR 1.95, 95% CI 1.15–3.31), but not with CRC (OR 0.98, 95% CI 0.63–1.50) [71], as reported in an old case-control study [36]. Interestingly, a sub-analysis showed that diverticulosis did not increase the risk of adenomas (OR 1.34, 95% CI 0.87–2.06) in patients who underwent screening colonoscopy [71]. The latter result might be related by the younger age of patients undergoing their first endoscopic CRC-screening. However, this work shows some limitations, such as the inclusion of cross-sectional studies (in which the causal relationship is difficult to assess), studies in which polyps were defined morphologically regardless of histologic evaluation, and the inclusions of data reported only in the form of abstracts.

In contrast with the results of the aforementioned meta-analysis [71], a cross-sectional study involving 2310 patients by Rondagh et al. reported that the association of diverticulosis with colorectal polyps was influenced by patient age: only below the age of 60 was the prevalence of colorectal polyps significantly higher in patients with diverticulosis than in those without diverticulosis [1]. An association with the female gender has also been described [41].

In the present study, all subtypes of polyps have been considered. The majority of all polyps were found in the distal colon. In particular, inflammatory polyps were only found in the distal colon. Adenomas accounted for 60% of all polyps. A significant increase in relative risk (RR 2.81, 95% CI 2.27–3.47, *p* < 0.0001) of colorectal adenoma and cancer in patients with colonic diverticula was found. At multivariate analysis, of the demographic and clinical parameters considered, only diverticula resulted in a significant association with colorectal adenomas and cancer (OR 3.86, 95% CI 2.90–5.14, *p* < 0.0001).

The location of inflammatory polyps in the distal colon confirms literature data, probably due to the presence of a mucosal inflammation of this colonic tract more commonly involved by diverticula [25,72,73,74,75]. Inflammatory polyps develop as a result of an exuberant mucosal regeneration and repair of mucosal inflammatory lesions. The inflammatory polyps are found not only in IBD but also in infectious colitis, ischemic colitis, necrotizing enterocolitis, graft versus host disease, and colonic anastomosis. In non-IBD colitis, inflammatory polyps do not represent a risk factor for CRC.

The association between colonic diverticula and colorectal adenomas and cancer could be related to the epithelial proliferation of colonic mucosa. An upward shifting of cellular proliferation in the colonic mucosa of patients with diverticula was demonstrated and compared with age-matched controls. The cell proliferation index in patients with asymptomatic diverticulosis is three-fold higher than that of healthy controls [76]. It is not clear whether this abnormal epithelial proliferation in colonic diverticula could be responsible for the development of inflammatory polyps, as well as colorectal adenomas and CRC.

In the cohort of patients considered in this study, differences in the frequency of polyps and cancer in patients with diverticulosis, SUDD, SCAD, and diverticulitis have not been detected.

Pathophysiology of diverticulosis is multifactorial, involving genetic predisposition, thickening of the circular and longitudinal muscle wall, elastosis, weakening of the colonic wall, altered neuromuscular activity, environmental factors, age, alteration of microbiota, and inflammation [77,78]. Environmental factors, age, microbiota, and inflammation are also implicated in the pathogenesis of colorectal carcinogenesis [24,79,80,81,82,83].

It has been reported that patients with DD, but also patients with asymptomatic diverticulosis, may display microscopic or macroscopic mucosal signs of mucosal inflammation [25,72]. The relation between chronic inflammation and cancer is well known. Patients with IBD have a significantly higher risk of developing CRC than unaffected subjects, a risk that increases with the extension and duration of the disease [82,83,84]. Furthermore, the epithelial hyper-proliferation of colon crypts, observed both in DD and asymptomatic diverticulosis [85], could be a potential cellular mechanism, leading to a possible increased risk of developing adenomas and CRC in patients with colonic diverticula. It is important to assess whether this increased risk of colorectal adenomas and cancer is linkable to colonic diverticula in general or only to chronic diverticulitis or diverticula with specific genetic and epigenetic variants.

On the other hand, some studies have hypothesized a protective effect of diverticula on CRC development [44]. It should also be noted that not all forms of chronic inflammation of the colon are associated with an increased risk of CRC. Recently, it was reported that microscopic colitis, chronic colitis with the same incidence and prevalence as IBD, appeared to show a reduced risk of adenomas and CRC [86,87].

However, our study shows, some limitations. As this was a cross-sectional study, it was not possible to establish the causal relationship between DD and neoplasm, as well as the future risk of developing CRC in patients with diverticulosis.

Moreover, the indication of a surveillance colonoscopy for a history of polyps was largely self-reported by the patient. Similarly, familial history of CRC and colonic diverticula were also self-reported by patients.

An additional possible limitation is represented by the non-systematic use of validated scales, reporting the extent of diverticulosis and inflammation, or the Paris classification, as the aim of the study focused on histologic findings.

Finally, results involving patients with cancer may be hampered by the limited number of CRCs detected in our study.

However, with the aforementioned limitations, the importance of our study derives not only from being a prospective study including a homogenous population of Caucasian patients but also from a detailed endoscopic localization of diverticula, polyps, and cancers in the various tracts of the large bowel and a histological definition of the various types of polyps and cancer detected.

## 5. Conclusions

Diverticula and adenomas are frequently found during colonoscopies in adult and elderly patients; however, the evidence of the association between these two conditions is still conflicting.

In our cohort of Caucasic patients, colonic diverticula and colorectal polyps resulted in a significant association, more so for colorectal adenomas and cancer. This finding suggests that patients with colonic diverticula, harboring a higher risk for dysplastic lesions, could require more vigilant endoscopic surveillance than the general population.

However, the present study was not designed to assess follow-up risk. Thus, further prospective large-scale population studies with a long follow-up are needed to corroborate the evidence supporting the relationship between colonic diverticula and colorectal adenomas and CRC, as well as to determine the best time interval between endoscopic examinations.

## Figures and Tables

**Figure 1 medicina-57-00108-f001:**
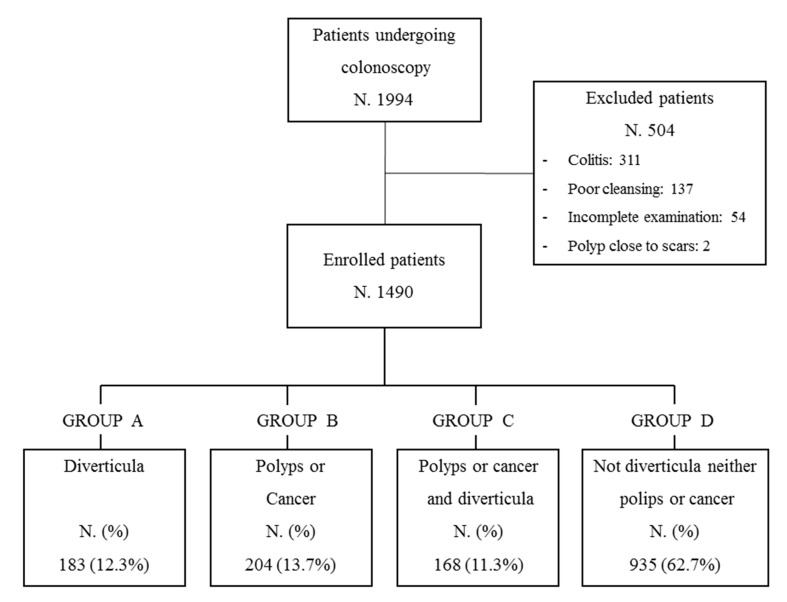
Study population.

**Figure 2 medicina-57-00108-f002:**
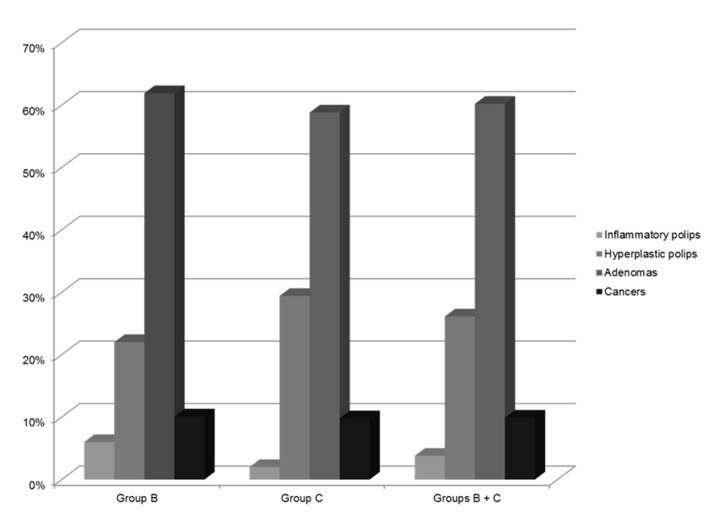
Histological distribution of all polyps and cancers found in Group B (patients with only polyps or cancer), in Group C (patients with both diverticula and polyps or cancers), and Group B + Group C.

**Table 1 medicina-57-00108-t001:** Demographic and clinical characteristics.

	GROUP A	GROUP B	GROUP C	GROUP D
	Diverticula	Polyps or Cancer	Polyps or Cancer and Diverticula	No Diverticula, Polyps, or Cancer
	(N.183)	(N.204)	(N.168)	(N.935)
Age (Years)				
Median (Range)	68 (27–92)	63 (21–93)	68 (47–92)	65 (29–90)
Sex				
Female N (%)	88 (48%)	95 (47%)	64 (38%)	410 (44%)
Male N (%)	95 (52%)	109 (53%)	104 (62%)	525 (56%)
Family history of CRC				
Negative N (%)	166 (90.71%)	152 (74.51%)	143 (85.12%)	783 (83.74%)
Positive N (%)	17 (9.29%)	52 (25.49%)	25 (14.88%)	152 (16.26%)
Family history of DD				
Negative N (%)	178 (97.27%)	202 (99.02)	160 (95.24%)	910 (97.33%)
Positive N (%)	5 (2.73%)	2 (0.98)	8 (4.76%)	25 (2.67%)

Colorectal cancer (CRC); diverticular disease (DD).

**Table 2 medicina-57-00108-t002:** Histological characteristics and localization of colorectal polyps and cancer in patients with diverticula of the colon (Group C).

Localization of Diverticula in the Colon Tracts	Histology			
	Inflammatory Polyps N (%) 10 (6.0%)	Hyperplastic Polyps N (%) 37 (22.0%)	Adenomas N (%) 104 (61.9%)	Cancer N (%) 17 (10.1%)
Proximal colon	0	2 (5.4%)	2 (1.9%)	2 (11.8%)
Distal colon	10 (100%)	29 (78.4%)	70 (67.3%)	15 (88.2%)
Proximal and distal colon	0	6 (16.2%)	32 (30.8%)	0

**Table 3 medicina-57-00108-t003:** The relative risk of the association of colorectal polyps and cancer with diverticula of the colon.

Polyps and Cancers	Diverticula		Total
	Absent	Present	
Absent	935	183	1118 (75.0%)
Present	204	168	372 (25.0%)
Total	1139	351	1490
Relative risk 2.67			
95% CI 2.27 to 3.15			
Significance level *p* < 0.0001			

**Table 4 medicina-57-00108-t004:** Relative risk of the association of colorectal adenoma and cancer with diverticula of the colon.

Adenoma and Cancers	Diverticula		Total
	Absent	Present	
Absent	999	230	1229 (82.5%)
Present	140	121	261 (17.5%)
Total	1139	351	1490
Relative risk 2.81			
95% CI 2.27 to 3.47			
Significance level *p* < 0.0001			

**Table 5 medicina-57-00108-t005:** Multivariate analysis with logistic regression and adjusted odds ratios (OR). Multiple logistic regression analyses including age, sex, family history for CRC, family history for diverticula, and presence of diverticula were carried out to investigate the independent association with colorectal polyps.

Variables	Odds Ratio	*p* Value
	(95% CI)	
Age		
per year	1.00 (0.99–1.01)	0.753
Sex		
female	1 (reference)	0.836
male	0.97 (0.74–1.28)	
Family history of CRC		
No	1 (reference)	0.162
Yes	1.32 (0.90–1.94)	
Family history of DD		
No	1 (reference)	0.467
Yes	0.71 (0.28–1.78)	
Diverticula		
No	1 (reference)	<0.0001
Yes	3.86 (2.90–5.14)	

**Table 6 medicina-57-00108-t006:** Studies evaluating the association between colonic diverticula and colorectal neoplasia.

Study	Study Design	N. Patients	Mean Age	Outcomes Assessed	Diagnosis	Association
McCallum A 1988 [35]	Retrospective	119	--	Association between diverticulosis and CRC	Barium enema	no
Morini S 1988 [36]	Case-control study	150	57.0	Association between diverticular disease, adenomas, and CRC	Colonoscopy	Yes for adenomas No for CRC
Stefansson T 1993 [37]	Retrospective	7159	--	Association between diverticulosis or diverticulitis and CRC	Nationwide register	Two-fold increase in the RR of left-sided CRC in patients with diverticulosis or diverticulitis.
Yusuf MA 2000 [38] ^a^	Retrospective	311	45.3	Association between diverticular disease and CRC	Colonoscopy	no
Loffeld R.J 2002 [39]	Retrospective	9086	52.0 without DD 69.0 with DD	Association between diverticulosis and CRC or polyps (histological type unspecified)	Colonoscopy	No for CRC yes for polyps
Morini S 2002 [26]	Prospective	630	66.8 with DD 61.5 without DD	Association between diverticular disease, adenomas, and CRC	Colonoscopy	Yes for adenomas No for CRC
Stefansson T 2004 [40]	Retrospective	7159	--	Association between sigmoid diverticulitis and increased risk of left-side CRC	Barium enema	yes
Kieff BJ 2004 [41]	Prospective	502	58.6	Association between diverticulosis and CRC	Colonoscopy CT scan surgery barium enema	Yes, only for women with extensive distal diverticulosis
Rajendra S 2005 [42]	Prospective	410	51.7	Association between diverticular disease and adenomas	Colonoscopy	yes
Soran A 2006 [43]	Retrospective	1561	67.0 with DD and CRC 61.0 only with DD	Association between diverticulosis and CRC	Surgery	no
Krones CJ 2006 [44]	Retrospective	1326	64.0	Association between diverticulosis or diverticulitis and CRC	Surgery	no
Choi CS 2007 [45]	Retrospective	2377	50.8	Association between diverticulosis and CRC	Colonoscopy	yes
Rangnekar AS 2007 [46] ^a^	Prospective	308	--	Association of diverticulosis and polyps	Colonoscopy	yes
Hirata T 2008 [47]	Retrospective	672	58.0	Association between diverticular disease and colonic polyps	Colonoscopy	yes
Meurs-Szojda MM 2008 [30]	Retrospective	4241	59.0	Association of diverticulosis, diverticulitis, polyps, and CRC	Colonoscopy	no
Hammoud I 2009 [48] ^a^	Retrospective	1668	--	Association of diverticulosis and polyps	Colonoscopy	no
Lee KM 2010 [49]	Prospective	1030	52.2	Association of diverticulosis and polyps	Colonoscopy	yes
Mazumder MK 2011 [50] ^a^	Retrospective	1000	57.3	Association of diverticulosis and polyps	Colonoscopy	yes
Neubauer K 2011 [51] ^a^	Retrospective	1776	52.5	Association between diverticulosis and colorectal adenomas and CRC	Colonoscopy	Yes for adenomas No for CRC
Rondagh EJ 2011 [1]	Prospective	2310	58.4	Association between diverticulosis and colorectal polyps (adenoma, serrated polyp, advanced CRC)	Colonoscopy	yes
Grandlund J 2011 [31]	Retrospective	41,037	75.0 F 73.0 M	Association between diverticular disease and CRC	CT-scan Colonoscopy	no
Szura M 2011 [52] ^a^	Retrospective	22,441	55.1	Association of diverticulosis and polyps	Colonoscopy	yes
Gohil VB 2012 [28]	Retrospective	300	57.0	Association between diverticulosis and adenoma detection rate	Colonoscopy	yes
Lee SJ 2012 [53]	Prospective	604	56.9	Association of diverticulosis and CRC	CT-colonography	no
Azzam N 2013 [54]	Retrospective	270	60.82	Association among colonic polyps, comorbidities, and diverticular disease	Colonoscopy	Yes for diverticulosis and adenomas
Parava P 2013 [55] ^a^	Retrospective	1077	57.0	Association of diverticulosis and polyps	Colonoscopy	yes
Muhammad A 2014 [29]	Retrospective	2223	63.0 with DD 59.0 without DD	Association between diverticulosis and colorectal polyps (adenomas and advanced adenomas)	Colonoscopy	Yes for adenomas
Lecleire S 2014 [56]	Retrospective	404	60.9 with DD 60.7 without DD	Association between acute diverticulitis and colorectal polyps	Colonoscopy	no
Meda S 2014 [57] ^a^	Retrospective	890	--	Association of diverticulosis and polyps	Colonoscopy	no
Shen H 2014 [58] ^a^	Prospective	1363	59.3	Association of diverticulosis and adenomas	Colonoscopy	no
Huang WY 2014 [59]	Retrospective	41,359	56.0	Association between diverticulosis or diverticulitis and CRC	Colonoscopy	no
Ashktorab H 2015 [27]	Retrospective	1986	57.0	Association between diverticulosis and pre-neoplastic colonic lesions (hyperplastic polyps and adenomas)	Colonoscopy	yes
Peery AF 2015 [32]	Prospective	624	56.0 with adenoma 53.0 without	Association between colonic diverticula and adenomas or advanced adenomas	Colonoscopy	no
Wang FW 2015 [60]	Prospective	1899	52.8	Association of diverticulosis and adenomas	Colonoscopy	yes
Wong ER 2016 [61]	Retrospective	2766	53.2	Association between colonic diverticula and CRC or polyps (not specified the histological type)	Colonoscopy	yes
Levine I 2017 [62]	Retrospective	589	63 with D 58 without D	Association between colonic diverticula and CRC or polyps	Colonoscopy	no
Shah R 2017 [63] ^a^	Retrospective	896	--	Association of diverticulosis and polyps	Colonoscopy	yes
Teixeira C 2017 [64]	Prospective	203	65.5	Association of diverticulosis and adenomas	Colonoscopy	no
Hong W 2018 [65]	Prospective	17,456	53.5 with D 49 without D	Association of diverticulosis or location of diverticulosis and adenoma or advanced adenoma	Colonoscopy	no
Pavão Borges V 2018 [66] ^a^	Retrospective	414	67.2	Association of diverticulosis and adenomas	Colonoscopy	no
Rodriguez-Castro K 2018 [67] ^a^	Prospective	25,962	67.2 with D 58.1 without D	Association between colonic diverticula and CRC or polyps	Colonoscopy	Yes for adenomas No for CRC
Schramm C 2018 [68]	Retrospective	4196	65.6 with D 62.0 without D	Association between colonic diverticula and CRC or polyps	Colonoscopy	Yes for adenomas No for CRC
Wang SF 2019 [69]	Retrospective	346,118	56	Association between colonic diverticula and CRC or polyps	Colonoscopy	yes
Tomaoglu K 2020 [70]	Retrospective	3496	51.3 M 53.3 F	Association between diverticulosis and CRC or polyps	Colonoscopy	Yes for adenomas No for CRC

CRC = Colorectal Cancer; DD = Diverticular Disease; RR = Relative Risk; CT = Computed Tomography; ^a^ = Data published only in Abstracts.

## Data Availability

No additional data are available.

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
