# Peer review of "Association of Colonic Diverticula with Colorectal Adenomas and Cancer"

_medicina, 2021, doi:10.3390/medicina57020108_

Round 1

Reviewer 1 Report

The authors conducted a prospective observational study to investigate the relationship between the colorectal diverticulum and colorectal tumors. The study was conducted in the 2 centers. Although the study is well designed, and the manuscript is professionally written. there are some concerns that must be addressed.

Major

  1. In the introduction section, the background of this study is well written, and the hypothesis that the inflammatory process caused by the diverticular disease may accelerate the malignant process seems rationale. However, as the authors described, similar studies have already been reported. What is the strong point of the present study compared with the previous reports or, what limitation is found in the previous reports? For example, if there was no reported prospective study, the present prospective study has scientific value.
  2. In 2.2. Colonoscopy and Histopathology section, the authors wrote that the patients with poor bowel preparation were excluded. The authors should describe the total number of patients included in the present study as well as the number of patients who were excluded. Moreover, regarding figure 1, the patient flow should be added to this figure. I mean X patients were included and Y patients were excluded (with reason), then finally 1490 patients were enrolled.
  3. In 2.1. Study Design and Population section, the authors should describe the way of research in this paragraph. Although that is shortly written in the statistical analysis section, it should be written as a methodology.
  4. The reason for the colonoscopy should be described in detail in the result section since the adenoma detection rate will change according to the target population.
  5. I think the location relationship of colorectal tumors and colorectal diverticulum should be described. If the local inflammatory process is important, the tumors would arise near the diverticula.
  6. In the discussion section, no description was found about the study limitation.

Minor

  1. In the result section, I am not sure whether the family history of diverticular disease is necessary. It is said that even collecting the correct colorectal cancer history is considered to be difficult then a universal screening of microsatellite instability is conducted to find the lynch syndrome. Considering that, I feel it is very difficult to obtain the correct family history of diverticular disease.
  2. About Table5 ‘CCR’ should be revised as ‘CRC’.
  3. I think that the discussion section is long, the authors should focus on the findings from the present study. I felt the review about the other literature is somehow long.

Author Response

Reviewer #1

We thank you very much for your suggestions and constructive comments, which have helped us to improve the article.

Major

1. In the introduction section, the background of this study is well written, and the hypothesis that the inflammatory process caused by the diverticular disease may accelerate the malignant process seems rationale. However, as the authors described, similar studies have already been reported. What is the strong point of the present study compared with the previous reports or, what limitation is found in the previous reports? For example, if there was no reported prospective study, the present prospective study has scientific value.

REPLY: In the Introduction section, we reported that most of the evidence available in literature derives from retrospective studies and the results from the few prospective cohorts are discordant, especially among different ethnic groups. These aspects were then clarified in more detail also in the Discussion section. Table 6 has been updated, including other new studies.

2. In 2.2. Colonoscopy and Histopathology section, the authors wrote that the patients with poor bowel preparation were excluded. The authors should describe the total number of patients included in the present study as well as the number of patients who were excluded. Moreover, regarding figure 1, the patient flow should be added to this figure. I mean X patients were included and Y patients were excluded (with reason), then finally 1490 patients were enrolled.

REPLY:

A new paragraph has been added in the Materials and Methods including the aforementioned new data which were requested and then reported also in the new Figure 1.

3. In 2.1. Study Design and Population section, the authors should describe the way of research in this paragraph. Although that is shortly written in the statistical analysis section, it should be written as a methodology.

REPLY: The above paragraph was simply changed to Study Population, leaving the description of the study design in the Statistical Analysis paragraph.

4. The reason for the colonoscopy should be described in detail in the result section since the adenoma detection rate will change according to the target population.

REPLY: The reasons for the colonoscopy were reported in detail in the section Study Population. We agree that the adenoma detection rate change according to the target population, but these data, unfortunately, were not available.

5. I think the location relationship of colorectal tumors and colorectal diverticulum should be described. If the local inflammatory process is important, the tumors would arise near the diverticula.

REPLY: In the Results, it was reported that most of the polyps and CRCs were located in the left colon where diverticula were also more frequently present. Almost all patients had no macroscopic signs of colitis associated with diverticula or signs of diverticulitis.

6. In the discussion section, no description was found about the study limitation.

REPLY: A new paragraph has been added in the Discussion section reporting the various limitations of our study.

Minor

1. In the result section, I am not sure whether the family history of diverticular disease is necessary. It is said that even collecting the correct colorectal cancer history is considered to be difficult then a universal screening of microsatellite instability is conducted to find the lynch syndrome. Considering that, I feel it is very difficult to obtain the correct family history of diverticular disease.

REPLY: We agree with these observations. However, in our cohort, the indication of a surveillance colonoscopy for a history of polyps was largely self-reported by the patient. Similarly, familial history of CRC and colonic diverticula were also self-reported by patients. This is one of the limitations of our study, as we pointed out in the Discussion section.

2. About Table5 ‘CCR’ should be revised as ‘CRC’.

REPLY: CCR was corrected as CRC

3. I think that the discussion section is long, the authors should focus on the findings from the present study. I felt the review about the other literature is somehow long.

REPLY: The Discussion section was significantly reduced.

Reviewer 2 Report

The authors prospectively estimated the relation between the colorectal polyps and diverticula.

Major comment

1. How many endoscopies were performed during the 3-year study period, how many of them were excluded for what reasons, and finally 1490 cases were included? The number of cases during the 3-year period seems rather small and may give the reader a sense of distrust, thus in order to clarify the quality of the study, it is desirable to describe the flow diagram.

2. Was it possible to observe the entire colon under favorable conditions in all of the included cases? Were there any cases in which total colonoscopy cannot be achieved, or in which preparation was poor?

3. Some of the eligible cases were surveillance cases, the findings of polyps could be related to a history of previous polypectomy. What was the breakdown of the purpose of the examination in each group?

4. This study evaluated the association between diverticula and polyps within one exam, and was not designed to assess follow-up or post-exam cancer risk. For this reason, the Conclusion is telling too-much which is unmentionable from this study design.

Minor comment

1. Page 3 Line 131, the number and percentage of the inflammatory polyp is incomplete.

Author Response

Reviewer #2

We thank you very much for your suggestions and constructive comments, which have helped us to improve the article.

Major comment

1. How many endoscopies were performed during the 3-year study period, how many of them were excluded for what reasons, and finally, 1490 cases were included? The number of cases during the 3-year period seems rather small and may give the reader a sense of distrust, thus in order to clarify the quality of the study, it is desirable to describe the flow diagram.

REPLY: A new paragraph has been added in the Materials and Methods including the aforementioned new data which were requested and then reported also in the new Figure 1.

2. Was it possible to observe the entire colon under favorable conditions in all of the included cases? Were there any cases in which total colonoscopy cannot be achieved, or in which preparation was poor?

REPLY:  These new data have been added in the Colonoscopy and Histopathology section and the new Figure 1.

3. Some of the eligible cases were surveillance cases, the findings of polyps could be related to a history of previous polypectomy. What was the breakdown of the purpose of the examination in each group?

REPLY: We agree that some surveillance cases could be related to a recurrence after a previous polypectomy. However, when polyps were detected close to the scar, patients were excluded. This clarification has been added in the Colonoscopy and Histopathology section and in the new Figure 1.

4. This study evaluated the association between diverticula and polyps within one exam, and was not designed to assess follow-up or post-exam cancer risk. For this reason, the Conclusion is telling too-much which is unmentionable from this study design.

REPLY: We agree with this observation. The lack of follow-up is one of the limitations of our study that we pointed out in the Discussion section.

Minor comment

1. Page 3 Line 131, the number and percentage of the inflammatory polyp is incomplete.

REPLY: This number was corrected.

Round 2

Reviewer 1 Report

The authors revised their manuscript well. Further studies are necessary to investigate the causal relationship between colonic diverticula and clonic neoplasms.

Author Response

We thank you very much for your suggestions and constructive comments.

The authors revised their manuscript well. Further studies are necessary to investigate the causal relationship between colonic diverticula and colonic neoplasms.

REPLY: In the conclusions of the manuscript, we stressed the need for further studies to increase the evidence of the causal relationship between colonic diverticula and colorectal neoplasms.